# Bone Metastasis in Bladder Cancer

**DOI:** 10.3390/jpm13010054

**Published:** 2022-12-27

**Authors:** Lei Yi, Kai Ai, Xurui Li, Zhihong Li, Yuan Li

**Affiliations:** 1Department of Urology, Second Xiangya Hospital, Central South University, Changsha 410017, China; 2Department of Orthopedics, Second Xiangya Hospital, Central South University, Changsha 410017, China

**Keywords:** BCa, bone metastasis, bone microenvironment, preclinical model, management

## Abstract

Bladder cancer (BCa) is the 10th most common and 13th most deadly malignancy worldwide. About 5% of BCa patients present initially with metastatic disease, with bone being the most diagnosed site for distant metastasis. The overall one-year survival of patients with BCa is 84%, whereas it is only 21% in patients with bone metastasis (BM). Metastasis of BCa cells to bone occurs by epithelial-to-mesenchymal transition, angiogenesis, intravasation, extravasation, and interactions with the bone microenvironment. However, the mechanism of BCa metastasis to the bone is not completely understood; it needs a further preclinical model to completely explain the process. As different imaging mechanisms, PET-CT cannot replace a radionuclide bone scan or an MRI for diagnosing BM. The management of BCa patients with BM includes chemotherapy, immunotherapy, targeted therapy, antibody-drug conjugates, bisphosphonates, denosumab, radioisotopes, and surgery. The objective of these treatments is to inhibit disease progression, improve overall survival, reduce skeletal-related events, relieve pain, and improve the quality of life of patients.

## 1. Introduction

Bladder cancer (BCa) is the 10th most common and 13th most deadly malignancy worldwide [1]. BCa is a heterogeneous disease, with approximately 75% of lesions being non-muscle-invasive, while 25% are muscle-invasive or metastatic [2,3]. Elucidation of the biology of BCa has essentially altered how the condition is diagnosed and treated. Despite timely treatment, muscle-invasive BCa (MIBC) can progress to metastatic BCa. Apart from lymph nodes, bone is the most common distant metastasis site for BCa. Cisplatin-containing chemotherapy is the current standard therapy for the treatment of metastatic BCa, although the one-year survival rate of patients with bone metastasis (BM) is still relatively low.

The understanding of the mechanisms of BM development has progressed significantly in recent years, resulting in numerous treatment options for patients. External beam radiotherapy (EBRT), bone-targeted agents, chemotherapy, biologically targeted therapy, immunotherapy, and systemically administered radioisotopes are important recommended treatment modalities. Surgery is useful in treating structural complications associated with bone destruction or nerve compression. The management of bone metastases is aimed primarily at inhibiting disease progression and relieving symptoms. A better understanding of the mechanism of BCa metastasis to the bone is therefore critical to developing treatments that may improve patient survival.

The objective of this article is to provide a review of the epidemiology, pathophysiology, molecular biology, preclinical models, diagnosis, and management of BM of BCa.

## 2. Epidemiology

In 2020 there were approximately 573,000 new cases of BCa throughout the world, which resulted in 213,000 deaths [1]. Previous studies have shown that about 5% of BCa patients present initially with metastatic disease, with bone being the most common site for distant metastasis. There is evidence that BM occurs in 30–40% of patients with metastatic BCa [4,5,6].

Previous studies have shown that BCa patients are possibly to be diagnosed with BM when they present with a series of factors, including higher primary tumor (T) stage, higher regional lymph node (N) stage, aged between 41 to 60 years, black race, poor tumor differentiated grade, squamous cell carcinoma (SCC), primary tumor site in the bladder neck, and the presence of lung, liver, and brain metastases [7].

A population-based study showed that the overall one-year survival of patients with BCa was 84% but was only 21% in patients with BM. The median overall survival time of 963 patients with BM was only 4 months (95% CI, 3.55–4.45 months) [8]. Other studies reported a longer overall survival time of BC patients with BM, particularly in those receiving combination therapy based on cisplatin [9,10,11].

## 3. Mechanisms/Pathophysiology

The vertebral veins with their rich, valveless ramifications and connections provide a direct potential metastatic route linking the pelvis to the spine [12,13,14]. The molecular mechanisms and pathophysiology of BM in BCa have received little attention. The metastatic process involves interaction with the tumor microenvironment and genetic alterations in cancer cells [15,16,17,18]. The process of the metastatic cascade involves several steps that include the separation of the tumor cells from the epithelial collective, degradation of the surrounding matrix, local invasion of the tumor, intravasation, survival of carcinoma cells in the circulation, chemotactic attraction and homing of cancer cells to the bone microenvironment, and reciprocal interactions with local immune cells and stromal cells [19,20,21].

### 3.1. Epithelial-to-Mesenchymal Transition

The first step in metastases necessitates a transfer of cancer cells from the primary tumor and their entry into the systemic circulation. Carcinoma cells accomplish this through the epithelial-to-mesenchymal transition (EMT) [22].

Cell-to-cell adhesion in the epithelium is accomplished by numerous junctions such as adherens and tight junctions composed of protein complexes of cell adhesion molecules [23]. E-cadherin works via homophilic binding interactions of the calcium-dependent extracellular binding domain of cells, while β-catenin is an adherence junction protein that anchors the actin cytoskeleton through its intracellular domain [24,25]. E-cadherin is crucial to the maintenance and differentiation of the epithelium phenotype in both normal and pathologic EMT [21]. By combining two E-boxes in the promoter region of E-cadherin, the transcription factors Snail1, Snail2, and Twist inhibit the expression of the promoter and drive this inductive morphogenesis [26].

EMT enables changes in epithelial-type cancer cells to perform mesenchymal traits, such as a loss of intercellular adhesion proteins between cells [27]. During EMT, the expression levels of mesenchymal markers, such as Slug, Snail, Zeb1, N-cadherin, and vimentin, are increased, while epithelial markers, such as E-cadherin, are reduced [28]. By commandeering developmental EMT pathways, epithelial cancer cells obtain the capability of migration, invasion, and metastasis resistance to chemotherapy [26,29].

In order to degrade the extracellular matrix and get away from tumor stroma, carcinoma cells also produce extracellular proteolytic enzymes [30]. After dissolving the extracellular matrix, the tumor cells invade local and surrounding tissues [30].

### 3.2. Angiogenesis, Intravasation and Extravasation

Angiogenesis is a precondition for tumor development and metastasis, and it has been suggested that BCa produces high levels of several stimulatory factors, such as vascular endothelial growth factor (VEGF) and fibroblast growth factor (FGF). Several studies found that endostatin reduces BCa development by reducing VEGF expression and inducing apoptosis [31,32]. By interacting with the vascular endothelium following promotion by cellular adhesion factors, such as integrin B1, members of the immunoglobulin and selectin families cause cancer cells to intravasate into circulation and extravasate into bone [33,34,35].

### 3.3. Metastases to Bone

BM in BCa is predominantly osteolytic and leads to bone destruction, followed by osteoblastic activity that results in the formation of new bone or mixed lesions containing both elements [10,36].

#### 3.3.1. Bone Microenvironment

After extravasation, the BCa cells encounter the bone microenvironment, which contains many cell compartments, including bone marrow stromal cells, immune cells, mesenchymal stem cells, fibroblasts, osteoblasts, osteoclasts, bone marrow endothelial cells, adipocytes, and hematopoietic cells [30,37,38].

#### 3.3.2. Osteoclasts

Osteoclasts are multinucleated cells responsible for bone resorption that originate from monocyte-macrophage and, when activated, release cathepsin K [39]. Growth factor colony-stimulating factor-1 (CSF-1) and the receptor activator of nuclear factor KB ligand (RANKL) are responsible for the maturity of osteoclasts. RANKL is a member of the tumor necrosis factor (TNF) family and is located in the membrane surface of both stromal cells and osteoblasts, and it is secreted by active T cells. Prostaglandins, parathyroid hormone, and 1,25-dihydroxyvitamin D3 have been shown to increase the expression level of RANKL [40,41]. RANKL promotes the formation of osteoclasts by combining the RANK receptor on osteoclast precursors, which then activates the nuclear factor-κB and Jun N-terminal kinase pathways. Osteoblasts are also activated by macrophage CSF (M-CSF) secreted by osteoblasts and stromal cells. Osteoprotegerin (OPG) secreted by osteoblasts is a decoy receptor for RANKL [42]. Universally, the ratio of RANKL to OPG controls the level of osteoclast maturity and activity [43].

#### 3.3.3. Osteoblasts

Osteoblast cells are originated from mesenchymal stem cells and are responsible for the deposition of collagen and mineralized calcium phosphate that maintains the bone structure and the synthesis of new bone [44]. The growth and differentiation of osteoblasts are regulated by several factors, such as platelet-derived growth factor (PDGF), bone morphogenetic proteins (BMPs), fibroblast growth factor (FGF), transforming growth factor-β (TGF-β), and Wnt protein [45,46]. Osteoblasts secrete calcified matrix and osteocalcin when they mature and finally become osteocytes [47]. By controlling the level of RANKL, M-CSF, OPG, and sclerostin, osteoblasts promote or inhibit the development of osteoclasts [30].

#### 3.3.4. Endothelial Cells

The bone microenvironment also includes endothelial cells, which are involved in BM. Endothelial cells express E-selectin, P-selectin, intercellular adhesion molecule A, and vascular adhesion molecule 1, which increased the adhesion of circulating tumor cells when they pass through the bone marrow [48]. Endothelial cells also facilitate BM by promoting cell dormancy and neovascularization associated with tumor growth development [30].

#### 3.3.5. Immune Cells

The bone marrow is a reservoir for macrophages, T cells, dendritic cells, and myeloid-derived cells [49]. B cells and T cells produce OPG and RANKL and regulate osteoclastogenesis [50,51,52,53]. B cells induce osteoclastogenesis under normal conditions whereas they inhibit this process under pro-inflammatory conditions [54]. T cells also regulate bone loss through a different pathway, such as stimulating osteoclastogenesis induced by TNFα and IL-17 [55,56,57]. Regulatory T cells (Tregs) inhibit osteoclasts after recognizing an antigen of these cells [58]. In response to extracellular calcium levels, bone macrophages also enhance bone formation mediated by osteoblasts [59].

#### 3.3.6. Adipocytes

The bone marrow contains only a very small number of adipocytes during its early development, whereas it is composed of about 70% adipose tissue when a person is 25 years old [60]. It has been reported that adipocytes provide energy to carcinoma cells in the bone marrow [30]. A study also found adipocytes in bone marrow promoted bone proliferation, tropism, and cancer cell survival by secreting factors such as TNF-α, vascular cell adhesion molecule 1 (VCAM-1), adiponectin, IL-1B, and IL-6 [60,61].

#### 3.3.7. Osteolytic Metastasis

Osteoclast cells induce bone to release growth factors and calcium. Bone loss will occur when osteoclast numbers are increased without regulation. RANKL and M-CSF which predominately originate from osteocytes, osteoblasts, and immune cells induce osteoclast differentiation [51,62]. Several factors, such as IL-6, IL-11, soluble intercellular adhesion molecule-1 (ICAM-1), parathyroid hormone-related peptide (PTHrP), and macrophage-stimulating protein (MSP), also directly promote osteoclast growth and activation or indirectly by activating RANKL secreted by other cells [63,64,65,66,67,68]. Interestingly, carcinoma cells also produce numerous factors that increase the number of osteoclasts or RANKL secreted by other cells, with this increased number of osteoclasts resulting in the destruction of bone rather than increasing the number of cancer cells [69]. Cancer cells produce cell surface receptors, chemokine receptors, and cell adhesion molecules, which result in attaching them to bone marrow stromal cells and bone matrix, which in turn induces angiogenic factors and bone-resorbing factors that promote tumor growth in bone [41]. Bone resorption induced by BCa cells also produces growth factors preserved in bone, such as transforming growth factor β (TGF-β), activin, FGF, PDGF, and insulin-like growth factor (IGF), which become activated and induce cancer cell development and further bone osteoclasia [70]. A vicious circle is therefore generated in which the growth of a tumor induces the destruction of bone, which in turn induces tumor growth.

#### 3.3.8. Osteoblastic Metastasis

The mechanism of bone formation of osteoblastic metastases is not completely understood. A previous study showed the factors that stimulated osteoblast growth and differentiation included systemic hormones, such as PTH, and factors were produced in the bone microenvironment. These included endothelin-1 (ET-1) [71], PDGF, growth differentiation factor 15 (GDF15), and bone morphogenic proteins (BMPs) [72], which belong to the TGFβ superfamily that induces osteoblast differentiation by activating the transcription factor, Runx-2 [73]. One theory regarding osteoblastic metastasis that also involves osteolysis is the bone disruption that results in factors being released from the bone matrix, which causes carcinoma cells to grow and survive and establish bone metastases [74]. Osteoblastic metastasis may also involve a vicious circle in which cancer cells promote osteoblasts and bone formation, which release growth factors required by carcinoma cells to grow and survive.

### 3.4. Other Mechanisms

As we discussed above (Figure 1), the mechanism of BCa metastasis to the bone is still not completely understood. There is evidence that Med19 may be a factor that participates in the invasiveness of BCa metastases to the bone, possibly via BMP-2 [75]. Other studies have also suggested that PI3K/Akt in the GSK3β/β-catenin pathway may regulate cell colonization of BCa metastasis to the bone [76]. Alternatively, the interaction between BMP-2 and TNF-α may induce local invasion and distant metastasis of BCa metastasis to the bone [77]. However, all these studies were superficial, and further studies are therefore required to validate the findings.

The BCa cells encounter the bone microenvironment after extravasation. The bone microenvironment includes various cells which promote the metastatic invasion of bladder cancer cells to the bone directly or by secreting various factors. There is a vicious cycle where the growth of the tumor induces the destruction of bone or bone formation, which in turn induces tumor growth. PTH, parathyroid hormone; ET-1, Endothelin-1; GDF-15, growth differentiation factor 15; PDGF, platelet-derived growth factor; BMPs, bone morphogenetic proteins; FGF, fibroblast growth factors; TGF-β, transforming growth factor-β; ICAM, intercellular adhesion molecule; VCAM, vascular cell adhesion molecule; IGF, insulin-like growth factor; M-CSF, macrophage- colony-stimulating factor; RANKL, receptor activator of nuclear factor KB ligand; NF-κB, Nuclear factor kappa-B; JNK, c-Jun N-terminal kinase; OPG, Osteoprotegerin.

## 4. Preclinical Model

A preclinical model is crucial for determining the mechanism of BCa metastases to bone. Several models have been used to study BCa growth and BM. Cancer cells can be implanted intravesical via the urethra, which may result in invasive orthotopic growth and metastases to lymph nodes and even distant organs [78,79]. A study also constructed an orthotopic xenograft model by making an incision in the lower abdomen, followed by injection of UM-UC-3 cells into the bladder muscle layer resulting in metastases to the liver and lung [28]. However, no study has demonstrated orthotopically inoculated cancer cells metastasis to bone. BCa metastases can also be researched through inoculation of BCa cells into the blood circulation through the lateral tail vein or the left cardiac ventricle of rodents (BM model) [80,81]. This model bypasses the invasion and intravasation of a tumor but can be used to examine the colonization of cancer cells to distant organs. Cancer cells can also be injected to the bone surface, marrow cavity, or periosteal surface to create a model of BM. However, this model may not mimic the metastatic process and therefore systemic or orthotopic inoculation of cancer cells may be the preferred option [82].

Molecular imaging modalities, such as magnetic resonance imaging (MRI), ultrasound, and optical imaging (bioluminescence or fluorescence), are used to assess tumor growth and metastasis [83,84,85,86]. Whole-body optical imaging (BLI) is another sensitive molecular imaging technique used to study tumor growth in animals by measuring viable cells in real-time.

Initial preclinical bladder cancer models that used BLI of UM-UC-3 cells with a first-generation firefly luciferase found no metastatic lesions from the orthotopic sites [81,86,87,88,89]. This may have been due to the low expression of the luciferase reporter construct. A later study re-engineered a bioluminescent indicator firefly luciferase to increase its expression level in mammalian cells by optimizing the codon and removing transcription-factor binding sites, which inhibited the expression of the construct (termed luciferase 2) [90]. Further studies constructed a human bladder cancer cell line with a luciferase signal (UM-UC-3luc2) by using a ubiquitously active CAG promoter coupled to luciferase 2 [91]. A subsequent study established an orthotopic model of metastasis by inoculating UM-UC-3luc2 cells into the bladder of thymic nude mice and created a BM model by inoculating these cells into the left cardiac ventricle. Bioluminescent imaging of human firefly luciferase 2–positive BCa in the mice was then performed using the IVIS100 imaging system. One week after inoculation, the study found bioluminescent signals which indicate micrometastatic deposits, and then bone metastases were found to be growing exponentially [90].

## 5. Diagnosis

The primary site of BM in BCa is the spine, followed by the pelvis, ribs, skull, femur, and the proximal end of the humerus. Other rare sites are also reported, such as the tibia [92]. Bone pain, such as acute pain caused by pathological fractures, is a frequent type of BCa-related pain usually involving the spine. Despite timely radical cystectomy, about 50% of MIBC patients progress to metastatic BCa. Several cases also reported NMIBC metastasis to bone [93,94].

### 5.1. Skeletal-Related Events (SREs)

BM in BCa usually leads to SREs which, including pathological fracture, spinal cord compression, hypercalcemia, and radiotherapy, demand to relieve bone pain or decrease bone damage and surgery for the treatment or prevention of fractures [10,95]. SREs may result in immobilization, poor quality of life, loss of independence, and reduced survival in BCa patients [96].

### 5.2. Imaging

Osteolytic metastasis needs to be greater than 1 cm in size and has destroyed over 30% of bone density in order to be observed by radiographs. Plain films are sufficiently sensitive to observe compression fractures of the spine and osteolytic metastases but are not reliable for identifying osteoblastic metastases [97].

Bone scintigraphy detects BM by osteoblast activity, increasing the accumulation of nuclear tracers such as technetium-99 methylene diphosphonate at sites of bone formation, independent of whether the sites are lytic or sclerotic [97,98]. Bone scans are highly sensitive and are the most common method used to investigate patients with clinically suspected BM but are not sufficiently specific to diagnose these lesions in BCa.

Computed tomography (CT) and magnetic resonance imaging (MRI) have also been shown to be accurate for detecting BM and can be used when the results of plain films or bone scans are in question [99], especially for differentiation between benign and malignant of single or few lesions. CT scans are sensitive and specific and are better for evaluating bone quality, bone destruction, calcified tumor matrix, and cortical erosions [100], whereas MRI scans are better for demonstrating marrow replacement and skip lesions, quantifying edema, and evaluating neurovascular involvement [101].

Positron emission tomography (PET)-CT is an alternative imaging mechanism but cannot replace radionuclide bone scans and MRI for diagnosing BM. PET can be used to assess early tumor formation in marrow and tumor metabolism and provide quantitative metabolic information. It is superior in differentiating between benign and malignant bony lesions. It can detect metastasis in rare regions but cannot provide exact anatomical locations.

### 5.3. Biopsy

When bone lesions identified during imaging are accompanied by definite metastases at other body sites, the lesions tissue biopsy is usually not necessary. However, histological confirmation of bone lesions tissue is recommended in patients with either a small number of bone lesions or equivocal imaging tests [102].

### 5.4. Biochemical Markers

Osteoblast activity is related to the serum levels of osteocalcin, type I procollagen C-propeptide, bone-specific alkaline phosphatase, while osteoclast activity is related to the serum levels of C-terminal telopeptide of type I collagen and tartrate-resistant acid phosphatase and the urinary levels of type I collagen cross-linked N-telopeptides [103]. However, the use of these markers to detect BM in BCa requires further study. Tumor markers of BCa are also not sufficiently sensitive to monitor BM [104], with alkaline phosphatase used more often in clinical practice to predict or monitor BM in BCa [3].

## 6. Management

There are three main principles of management strategies for BCa with BM: 1. inhibiting or killing the BCa cells which may extend the survival of the patient [105]. This strategy includes killing the cancer cells directly and inhibiting the pathway of growth and metastasis of the tumor. 2. Focusing on the bone microenvironment. As discussed above, the interaction between BCa cells and the bone microenvironment creates a vicious circle and is crucial for BCa metastasis to bone. It is therefore useful to disrupt these interactions. 3. Palliative treatment of bone pain and SREs to enhance the quality of life of cancer patients.

### 6.1. Chemotherapy

#### 6.1.1. First-Line Chemotherapy for Metastatic Disease

The current line of therapy, as per NCCN guidelines for MIBC patients who went under radical cystectomy, is followed by adjuvant gemcitabine plus cisplatin chemotherapy (GC) followed by bone scintigraphy. Cisplatin-based combination chemotherapy includes methotrexate, vinblastine, doxorubicin, cisplatin (MVAC), and GC. A phase three randomized trial reported that patients in the MVAC group had higher rates of neutropenia, grade 3 and 4 mucositis, and neutropenic sepsis and fever than that observed in the GC group. This difference occurred despite equivalent oncologic outcomes in the two groups. GC is therefore generally preferred clinically, particularly for frail patients because it has a better adverse-effect profile than MVAC. The most significant side effect of cisplatin is nephrotoxicity, with its dose often adjusted according to the patient’s kidney function [106].

Carboplatin-based combined chemotherapy is used as a first-line treatment for metastatic BCa in patients with poor general health and impaired renal function, with carboplatin-gemcitabine being the first choice in cisplatin-ineligible patients [3].

Taxanes have also been reported to have modest anti-tumor activity in BCa. The NCCN guidelines recommend taxane-based combination chemotherapy as first-line chemotherapy for patients in whom cisplatin treatment is not suitable (2B), while the EAU guidelines recommend PCG as the first-line chemotherapy for those with metastatic BCa (A).

#### 6.1.2. Second-Line Chemotherapy and beyond for Metastatic Disease

There is no clear recommendation for the second-line treatment of metastatic BCa. Single drugs that are effective for BCa include platinum, taxon, methotrexate, bleomycin, pemetrexed, fluorouracil, and vinblastine. When first-line chemotherapy is not effective, drugs that have not been used in first-line chemotherapy from the above drugs can be used for second-line chemotherapy. For example, vinflunine is recommended in EAU guidelines for the therapy of advanced or metastatic BCa [107]. Immunotherapy Checkpoint pathways which regulate autoimmunity can be deprived by carcinoma cells to escape from immune response [108]. BCa is susceptible to immunotherapy, particularly with checkpoint inhibitors, such as monoclonal antibodies against programmed cell death-1 (PD-1), its ligand PD-L1, and cytotoxic T lymphocyte-associated protein 4 (CTLA-4) [107].

Since 2016, five checkpoint inhibitors have been given permission for the treatment of BCa. First-line immunotherapy is currently being researched, only pembrolizumab (anti-PD-1) and atezolizumab (anti-PD-L1) have been given permission as first-line treatments in cisplatin-ineligible patients and only in cases with positive PD-L1 status. Avelumab and durvalumab (anti-PD-L1), and nivolumab (anti-PD-1) have also been demonstrated to have clinical benefits in metastatic BCa and are approved for second-line treatment. Ipilimumab is a fully humanized anti-CTLA-4 IgG1 monoclonal antibody that is currently undergoing clinical trials [109].

### 6.2. Targeted Therapies and Antibody-Drug Conjugates

FGFR is a receptor tyrosine kinase that participates in cell growth, survival, and migration and is therefore a target in BCa, especially in luminal-subtype tumors [110].

Research in patients with locally advanced and metastatic disease who did not respond to prior therapy showed a 40% objective response rate with oral erdafitinib, a pan-FGFR inhibitor. This result led to erdafitinib being authorized by the FDA as a second-line treatment [111].

Antibody-drug conjugates utilize tumor proteins as targets for drug delivery [112]. Sacituzumab govitecan, an antibody-drug conjugate which links a topoisomerase inhibitor with an antibody against trophoblast cell surface marker 2 has been found to have a 29% overall response rate in patients whose tumor progressed with chemotherapy and immunotherapy.

### 6.3. Bisphosphonates

Bisphosphonates include nitrogen-containing molecules such as ibantdronate, pamidronate, alendronate, risedronate, zoledronate, and nitrogen-free molecules such as clodronate, tiludronate, and etidronate [113]. Bisphosphonates inhibit bone resorption by producing apoptosis of osteoclasts after they have been endocytosed. Non-nitrogen-containing bisphosphonates impair osteoclast activity by causing intracellular accumulation of cytotoxic non-hydrolysable ATP analogues [114]. In contrast, nitrogen-containing bisphosphonates prevent the prenylation of small GTPase signaling proteins by suppressing the mevalonate pathway, which results in the impaired capacity of osteoclasts to resorb bone [115]. Apoptosis of osteoclasts also inhibits the interaction between BCa cells and the bone microenvironment.

Previous studies reported that bisphosphonates prevent and delay BM in solid tumors [116]. Zoledronic acid has the best efficacy of the bisphosphonate molecules and is used widely in clinical practice. In a clinical trial including 40 patients with BCa metastasis to bone, zoledronic acid was shown to decrease the risk of SREs and relieved bone pain [117]. However, whether bisphosphonates improve the overall survival of patients with BCa metastasis to the bone remains controversial and requires further study. There are also adverse side effects with the usage of bisphosphonates, such as osteonecrosis of the jaw and hypocalcemia, and, therefore, patients receiving these agents should be supplemented with vitamin D and calcium [4].

### 6.4. Denosumab

Denosumab is a monoclonal IgG2 antibody found in humans, which functions by combining to the membrane-bound and DE loop region of RANKL, which is primarily produced by osteoblasts, inflammatory cells, and stromal cells and stimulates osteoclastic activity as we discussed above [118,119,120]. In this way, denosumab prevents bone resorption, bone pain, and hypercalcemia and may have an effect on cancer cells [121,122,123].

The FDA has approved denosumab as a study which showed that it prolonged the time to SRE events compared with that observed with zoledronic acid [124]. However, subsequent randomized, controlled trials investigating several solid tumors demonstrated that denosumab was not superior to zoledronate but rather was non-inferior [125]. Despite denosumab’s additional benefits on SREs, the overall survival and the time to disease progression of patients [98] treated with either zoledronate or denosumab showed no significant difference. Denosumab has been proven to inhibit osteoclast in several solid tumour bone metastases and was recommended for use in BCa patients with BM [4,126].

### 6.5. Clinical Use of Bisphosphonates and Denosumab

Treatment of bisphosphonates and denosumab should be used after diagnosis of BM in BCa and maintained throughout the course of the disease with sequential systemic treatments. For the reason that bisphosphonates attach to the bone surface, their duration of action is long and therefore zoledronate treatment can be reduced when the disease is in remission [127]. However, the half-life of denosumab is short and its discontinuation might increase the rate of SREs [128]. Therefore, adherence to monthly treatment of denosumab is recommended and when this treatment is discontinued, it is recommended to administer a potent bisphosphonate [129].

### 6.6. Radioisotopes

The radioisotopes used to treat BM are either alkaline earth metals or conjugated to ligands which deliver the radioisotope to the bone. Alkaline earth metals and calcium have the same electron valency and therefore alkaline earth metals become concentrated with calcium in regions of high bone turnover. Approved radioisotopes include β-emitters and α-emitters. Two β-emitters, stontium-89 and samarium-153 are used to treat bone pain induced by cancer but may also cause bone marrow suppression, which limits the use of chemotherapy [127]. Radium-223 is an α-emitter with preferential uptake into areas of increased bone formation, a characteristic that prolongs the time to SRE events and increases overall survival [130]. The rate of myelosuppression using radium-223 is small, with only mild thrombocytopenia being observed [131].

The advantage of radio-labeled tracers is that radiation can be delivered to the tumor more specifically than that achieved with external beam radiotherapy (EBRT), thereby avoiding the destruction of normal tissues. The ideal patients for radioisotope treatment are those with osteoblastic or mixed metastatic lesions which are multifocal and result in bone pain [98]. However, the curative effect of radioisotopes in BCa patients with BM requires further study.

### 6.7. External Beam Radiotherapy

Radiotherapy is a significant palliative treatment for BCa patients with BM. Radiotherapy is a non-invasive method for improving pain generally within 2–6 weeks of treatment. The ideal candidates for this treatment are patients with oligometastatic or a solitary BCa to the bone.

In addition to relieving pain, EBRT also promotes bone healing through its effects on the BCa. Approximately 80% of patients have a remission in bone pain. A single 8-Gy fraction of EBRT is equally efficient as multi-fractionated radiotherapy for relieving pain caused by BM without fractures or nerve entrapment [132]. Single-fraction radiotherapy also has fewer adverse events. The administration of antiemetics is recommended for patients who receive EBRT to the pelvis and dorso-lumbar spine, which may cause the oesophagus or other mucosal surfaces in the radiated area to become irritated. About 40% of patients will undergo aggravated pain before pain relief, although this can be prevented by the use of dexamethasone [133,134]. Stereotactic body radiotherapy (SBRT) utilizes 3D images to deliver accurate doses of EBRT to metastasis and minimize the disruption to normal tissues. Patients receiving SBRT have been shown to experience a more rapid and durable relief of pain than those receiving standard conformal EBRT [135].

### 6.8. Surgery

The survival benefits of surgery in the treatment of BCa with BM remain controversial. For solitary or oligometastasis and small lesions, resection of the metastatic lesions may be an effective treatment for patients with BCa. Palliative surgery is mainly carried out to relieve symptoms in BCa patients with BM.

The objective of surgical intervention for BM in BCa is mainly to enhance the quality of life of patients by preventing fractures or repairing a fracture, maintaining their functionality and mobility, managing spinal cord compression, and alleviating bone pain. An oligometastasis and small BM lesions should be excised completely with clear margins to prevent later recurrence and complications. Vertebroplasty or kyphoplasty can be used to treat pain caused by vertebral body fractures, usually relieving pain within 1–3 days after treatment [98].

### 6.9. Therapeutic Strategies of Bone Pain

As discussed above, zoledronic acid and denosumab were shown to relieve bone pain. Radiotherapy is a method for improving pain generally within 2–6 weeks of treatment. Vertebroplasty or kyphoplasty can be used to treat pain caused by vertebral body fractures, usually relieving pain within 1–3 days after treatment.

Opioids and certain non-opioid adjuvants are useful in relieving pain but also have a side effect that restricts their long-term administration. Currently, research is focused on pain-relieving medications with few side effects. Growth factors such as nerve growth factor (NGF), glial-derived neurotrophic factor (GDNF), and brain-derived growth factor (BDGF) participate in neural cell differentiation and maturation and their receptors are expressed within the spinal cord. A clinical trial found that an anti-NGF antibody, tanezumab, relieved pain more substantially in BM patients with higher baseline pain and lower baseline opioid use [136]. The transient receptor potential cation channel subfamily V member 1 (TRPV1) family of channels is located on unmyelinated C fibers and spinal nociceptive neurons that mediate pain transmission. A trial found TRPV1 antagonists have a potent analgesic effect [137].

## 7. Conclusions and Future Prospects

A better understanding of the mechanism of BCa metastases to bone can lead to more effective treatments for patients. Osteoblasts, osteoclasts, and other resident bone cells communicate during bone remodeling, responding to hormonal changes and mechanical loading. Metastatic BCa cells flourish in this microenvironment and stimulate osteoblasts, osteoclasts, and other resident bone cells to secrete factors that further promote BCa cell growth. However, it needs further studies that what interactions induce BCa cell dormancy versus those that promote BCa cell growth.

Present therapies for BM mainly are palliative, with few effects on long-term survival. Starting treatment before micrometastases progress may increase the time to progression and survival. Thus, it is a significant study that develops diagnostic techniques to detect early BM. Patients were treated with bone-targeted therapies such as bisphosphonates when BM was diagnosed. The future of bone-targeted therapies includes cytotoxic medications to BCa cells, small molecule inhibitors and neutralising antibodies that inhibit the homing of BCa cells to bone and the response of tumour in bone, and agents that change the bone microenvironment making it inhospitable to BCa cells. The future looks bright for promising treatments that can preserve bone function to inhibit bone metastases, and further study is needed to make use of the tumor-promoting cells and factors in the bone microenvironment to inhibit BM. No one therapy will abolish BCa metastasis to the bone but combined therapies that aim at individual pathways responsible for bone metastases will be efficient.

## Figures and Tables

**Figure 1 jpm-13-00054-f001:**
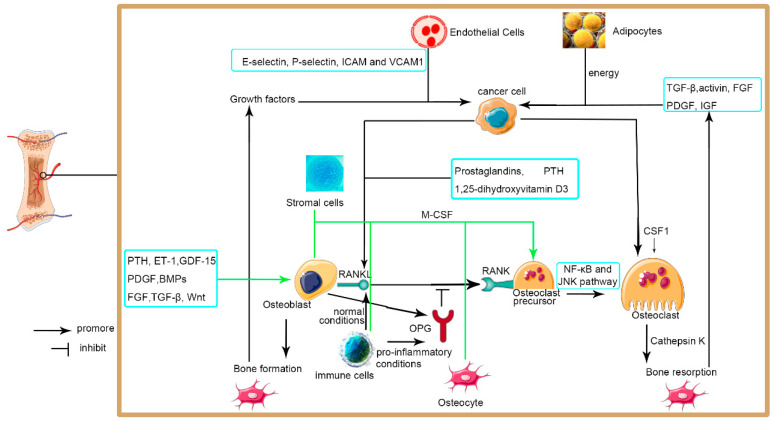
Metastatic invasion of cancer cells to the bone.

## Data Availability

Data sharing is not applicable to this article.

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
