# Peer review of "Bone Metastasis in Bladder Cancer"

_jpm, 2022, doi:10.3390/jpm13010054_

Round 1

Reviewer 1 Report

The objective of the manuscript “Bone Metastasis in Bladder Cancer” is to provide an evidence-based review of the epidemiology, pathophysiology, molecular biology, preclinical models, diagnosis, and management of bone metastases of bladder cancer. It is a well-structured review that provides all the pertinent information on the topic. Although this theme has been discussed in the past, it is always necessary to update it. 

As it appears in every manuscript published in this journal, I suggest that all reference numbers be placed in front of the point. In addition, the review should be suitable for publication after a final check of the English and the text.

Author Response

As it appears in every manuscript published in this journal, I suggest that all reference numbers be placed in front of the point. In addition, the review should be suitable for publication after a final check of the English and the text.

Response: Thank you very much for your suggestion. We have placed all reference numbers in front of the point and  revised the English and the text.

Reviewer 2 Report

The kind of review and selection must be explained ( time interval of research, relevant topics etc). It sounds as a narrative review without any other methodological issues.

The work is well focused on pathological development of BM in BC. Some other clinica data could be added as for example other mateastasis sites and their fisiopathology.

You shuould add the venous plexus linking pelvis to spine ( see references below)

PET role could be improved.

There is no need to consider surgery or EBRT for palliative treatment of MIBC in this work as you are focusing only on BM and not on bladder cancer management.

You could add a paragraph on pain management of BM symptoms.

Batson OV (1940) The function of the vertebral veins and their role in the spread of metastases. Ann Surg 112:138–149

The periprostatic venous plexus: an unusual source of fatal pulmonary embolism during corporoplasty.

Foschi N, Ragonese M, Grassi VM, De Matteis V, De-Giorgio F. Int J Legal Med. 2017 May;131(3):713-717.

Nathoo N, Caris EC, Wiener JA, Mendel E (2011) History of the vertebral venous plexus and the significant contributions of Breschet and Batson. Neurosurgery 69:1007–1014 

Author Response

The kind of review and selection must be explained ( time interval of research, relevant topics etc). It sounds as a narrative review without any other methodological issues.

Response: Thank you very much for your suggestion. We have revised it. We provided a review of the epidemiology, pathophysiology, molecular biology, preclinical models, diagnosis, and management of BM of BCa and focused on mechanisms/pathophysiology and therapy by 2022.

The work is well focused on pathological development of BM in BC. Some other clinica data could be added as for example other mateastasis sites and their fisiopathology.

Response: Thank you very much for your suggestion. We have added the data in our revised manuscript. The primary site of BM in BCa is the spine, followed by the pelvis, ribs, skull, femur, and the proximal end of the humerus. Other rare sites also be reported such as tibia.

You shuould add the venous plexus linking pelvis to spine (see references below)

Response: Thank you very much for your suggestion.  We have added it in our revised manuscript.

The primary site of BM in BCa is the spine, followed by the pelvis. The vertebral veins with their rich, valveless ramifications and connections provide a direct potential metastatic route linking pelvis to spine.

  1. Batson, O.V. THE FUNCTION OF THE VERTEBRAL VEINS AND THEIR ROLE IN THE SPREAD OF METASTASES. Ann Surg 1940, 112, 138-149, doi:10.1097/00000658-194007000-00016.
  2. Foschi, N.; Ragonese, M.; Grassi, V.M.; De Matteis, V.; De-Giorgio, F. The periprostatic venous plexus: an unusual source of fatal pulmonary embolism during corporoplasty. Int J Legal Med 2017, 131, 713-717, doi:10.1007/s00414-016-1519-9.
  3. Nathoo, N.; Caris, E.C.; Wiener, J.A.; Mendel, E. History of the vertebral venous plexus and the significant contributions of Breschet and Batson. Neurosurgery 2011, 69, 1007-1014; discussion 1014, doi:10.1227/NEU.0b013e3182274865.

PET role could be improved.

Response: Thank you very much for your suggestion.

PET can be used to assess early tumor formation in marrow and tumor metabolism and provide quantitative metabolic information. It is superior in differentiating between benign and malignant bony lesions.  It can detect metastasis in rare region but not provide exact anatomical location.

There is no need to consider surgery or EBRT for palliative treatment of MIBC in this work as you are focusing only on BM and not on bladder cancer management.

Response: Thank you very much for your suggestion. We have deleted it in our revised manuscript.

You could add a paragraph on pain management of BM symptoms

Response: Thank you very much for your suggestion. We have added a paragraph on therapeutic strategies of bone pain.

As discussed above, zoledronic acid and denosumab was shown to relieve bone pain. Radiotherapy is method for improving pain generally within 2–6 weeks of treatment. Vertebroplasty or kyphoplasty can be used to treat pain caused by vertebral body fractures, usually relieving pain within 1-3 days after treament.

Opioids and certain nonopioid adjuvants are useful in relieve pain but also have their side-effect that restrict their long-term administration. Currently research is focused on pain-relieving medications with little side effects. Growth factors such as nerve growth factor (NGF), glial-derived neurotrophic factor (GDNF), and brain-derived growth factor (BDGF) participate in neural cell differentiation and maturation and their receptors are expressed within the spinal cord. A clinical trial found anti-NGF antibody, tanezumab, relieve pain more substantial in BM patients with higher baseline pain and lower baseline opioid use[141]. The transient receptor potential cation channel subfamily V member 1 (TRPV1)family of channels is located on unmyelinated C fibers and spinal nociceptive neurons that mediate pain transmission.  A trial found TRPV1 antagonist have a potent analgesic effect[142].

Reviewer 3 Report

The bone metastasis has been observed with bladder cancer, however they are considerably less common when compared to prostate and breast cancer.

Author in the review mentioned in the abstract that “Several preclinical models having been proposed, all of which have flaws to completely explain the process” which is very offensive type of language.

In the introduction author mentioned that “Despite timely treatment, muscle-invasive BCa (MIBC) can progresses to metastatic BCa” on what basis author is clamming this sentence.

Author mentioned in the review article that cisplatin containing chemotherapy is the current standard therapy for treatment of metastatic BCa, which is not correct. The current line of therapy as per NCCN guideline for MIBC patients who went under radical cystectomy is followed by adjuvant gemcitabine plus cisplatin chemotherapy (GC) followed by bone scintigraphy.

In the abstract section and epidemiology there are several redundancies in the dialect.

Author in the manuscript escaped the important part for e.g., the common bone metastases sites for bladder carcinoma which includes pelvis and spine. Some paper reported in tibia also which is very rare in bladder cancer.

Altogether the review was lost the direction and author discussed much about the therapy option rather than discussing about bone metastasis in bladder. Author also did not mention about the unique case report where NMIBC turned to bone metastasis.  

Author Response

The bone metastasis has been observed with bladder cancer, however they are considerably less common when compared to prostate and breast cancer.

Author in the review mentioned in the abstract that “Several preclinical models having been proposed, all of which have flaws to completely explain the process” which is very offensive type of language.

Response: Thank you very much for your suggestion. We have revised it:

The mechanism of BCa metastasis to bone is not completely understood, it needs further preclinical model to completely explain the process.

In the introduction author mentioned that “Despite timely treatment, muscle-invasive BCa (MIBC) can progresses to metastatic BCa” on what basis author is clamming this sentence.

Response: Thank you very much for your suggestion.

Despite timely radical cystectomy, about 50% of the MIBC patients progresses to metastatic BCa.

Author mentioned in the review article that cisplatin containing chemotherapy is the current standard therapy for treatment of metastatic BCa, which is not correct. The current line of therapy as per NCCN guideline for MIBC patients who went under radical cystectomy is followed by adjuvant gemcitabine plus cisplatin chemotherapy (GC) followed by bone scintigraphy.

Response: Thank you very much for your suggestion. We have revised it in our manuscript.

The current line of therapy as per NCCN guideline for MIBC patients who went un-der radical cystectomy is followed by adjuvant gemcitabine plus cisplatin chemotherapy (GC) followed by bone scintigraphy.  Cisplatin-based combination chemotherapy includes methotrexate, vinblastine, doxorubicin, cisplatin (MVAC), and GC.  A phase three ran-domized trial reported that patients in the MVAC group had higher rates of neutropenia, grade 3 and 4 mucositis, and neutropenic sepsis and fever than that observed in the GC group. This difference occurred despite equivalent oncologic outcomes in the two groups. GC is therefore generally preferred clinically, particularly for frail patients because it has a better adverse-effect profile than MVAC.

In the abstract section and epidemiology there are several redundancies in the dialect.

Response: Thank you very much for your suggestion. We have revised it in our manuscript.

Author in the manuscript escaped the important part for e.g., the common bone metastases sites for bladder carcinoma which includes pelvis and spine. Some paper reported in tibia also which is very rare in bladder cancer.

Response: Thank you very much for your suggestion.  We have added the new data in our revised manuscript.

The primary site of BM in BCa is the spine, followed by the pelvis, ribs, skull, femur, and the proximal end of the humerus. The vertebral veins with their rich, valveless ramifications and connections provide a direct potential metastatic route linking pelvis to spine. Other rare sites also be reported such as tibia.  

Wheelock, M.C.; O'Conor, V.J., Jr. Metastasis in tibia from carcinoma of the urinary bladder. Q Bull Northwest Univ Med Sch 1953, 27, 111-113.

Altogether the review was lost the direction and author discussed much about the therapy option rather than discussing about bone metastasis in bladder. Author also did not mention about the unique case report where NMIBC turned to bone metastasis. 

Response: Thank you very much for your suggestion. We have added the data in our revised manuscript. Several cases reported NMIBC metastasis to bone such as:

            Hong, J.H. Early isolated bone metastases without local recurrence in non-muscle invasive bladder cancer. Int J Surg Case Rep 2015, 10, 41-44, doi:10.1016/j.ijscr.2015.03.029.

            Sasaki, Y.; Oi, H.; Oyama, T.; Kagawa, J.; Komori, M.; Senzaki, T.; Fukawa, T.; Takahashi, H.; Takemura, M.; Yamaguchi, K.; et al. [Non-muscle invasive bladder cancer with multiple bone metastasis without local invasion : a case report]. Hinyokika Kiyo 2013, 59, 669-672.

Round 2

Reviewer 1 Report

The manuscript has been improved, and it is suitable for publication. Congrats to the authors.

Reviewer 2 Report

Thank you for re-organize the paper and add relevant infomations on therapy and pathology models.

Reviewer 3 Report

The author address all my concerns and modified the manuscript as required. 

Just to remind the author to update more about the current line of therapy and NCCN guidelines.  I am just hoping that this article will be scientifically useful globally working in the area of bone metastasis.